# Development of AI-Based Tools for Power Generation Prediction

**Ana Paula Aravena-Cifuentes [1], Jose David Nuñez-Gonzalez [1], Andoni Elola [2] and Malinka Ivanova [3],***

1   Department of Applied Mathematics, University of the Basque Country (UPV/EHU), 20600 Eibar, Spain
2   Department of Electronic Technology, University of the Basque Country, 20600 Eibar, Spain;
    andoni.elola@ehu.eus
3   Department of Informatics, Faculty of Applied Mathematics and Informatics, Technical University of Sofia,
    1797 Sofia, Bulgaria
*   Correspondence: m_ivanova@tu-sofia.bg

**Abstract:** This study presents a model for predicting photovoltaic power generation based on meteorological, temporal and geographical variables, without using irradiance values, which have traditionally posed challenges and difficulties for accurate predictions. Validation methods and evaluation metrics are used to analyse four different approaches that vary in the distribution of the training and test database, and whether or not location-independent modelling is performed. The coefficient of determination, $R^2$, is used to measure the proportion of variation in photovoltaic power generation that can be explained by the model's variables, while $gCO_2eq$ represents the amount of $CO_2$ emissions equivalent to each unit of power generation. Both are used to compare model performance and environmental impact. The results show significant differences between the locations, with substantial improvements in some cases, while in others improvements are limited. The importance of customising the predictive model for each specific location is emphasised. Furthermore, it is concluded that environmental impact studies in model production are an additional step towards the creation of more sustainable and efficient models. Likewise, this research considers both the accuracy of solar energy predictions and the environmental impact of the computational resources used in the process, thereby promoting the responsible and sustainable progress of data science.

**Keywords:** energy; prediction; regression; r-squared

## 1. Introduction

Energy, or more specifically electricity, is one of the most significant pillars of society. Not only it is of vital importance to people's daily lives, but due to the growing population and economic growth its consumption will continue to increase substantially in the future [1,2]. The changing landscape of energy systems and the increasing dependence on electricity make it necessary to develop strategies to mitigate the impact of energy disruptions [3].

As global efforts to reduce greenhouse gas emissions and decarbonisation continue, renewable energy sources such as solar and wind power are being integrated into the energy systems faster than any other fuel in history [1,4].

Solar Photovoltaic energy has emerged in the last few decades as the most flourishing source of power generation [5]. Not only is it a clean and renewable energy, but it is also economically accessible with minimal maintenance.

Nevertheless, they have the disadvantage of high dependence on climatic factors, significant variability and high cost of energy storage. Hence, forecasting the generation of Photovoltaic (PV) installations for a given period of time can help to make optimal use of resources, allowing for reduced emissions, lower costs, safe operation and better integration into the grid [6,7].

Furthermore predicting solar energy generation offers intriguing prospects and simultaneous challenges as an accurate forecasts enables efficient grid integration and informed decision-making in energy trading and storage [8].

The prediction of PV power generation has been extensively studied in the literature using different approaches; generally a single location is used [9–12].

Previous generation values or the irradiance [9,13] at the time are typically used as the only or most important characteristics for predicting PV generation. The problem with this is the lack of a record of this data and its difficult accessibility. For this reason, a large number of studies concentrate on the prediction of irradiance [14–20].

In particular, irradiance is a parameter that, depending on the location, may be very accessible or not recorded at all. In addition, it requires precise instruments for its measurement, calibration and maintenance, which generates uncertainty about the reliability of the data.

In this study the prediction of the power of Photovoltaic installation was based on the selection of both geographical, seasonal and meteorological variables. Irradiance was not considered as a variable because of the problems associated with its use.

Machine learning approaches and techniques are studied with the aim of developing a model that is able to predict Photovoltaic generation. Building on previous work by Pasion et al. [21], an algorithm is developed that predicts the energy generated by Photovoltaic installations.

The general objectives of this work and their corresponding sub-objectives are defined as follows:

1. Reproduce the initial results of the baseline study

    1.1 Develop and apply the instantaneous prediction model proposed in the study by Pasion et al.

    1.2 Extend the state of the art.

2. Propose a different approach to improve the results on predicting the actual Photovoltaic generation.

    2.1 Explore and identify relevant Machine Learning algorithms aimed at improving the prediction performance.

    2.2 Propose different folding approaches and compare their results with the previous techniques.

This paper is structured as follows: Section 1 introduces the topic and its motivation. Section 2 outlines the context in which the work was carried out, along with the objectives and structure of the study. Section 3 presents the dataset and the methodology employed for its management, as well as the construction of models using different techniques. Section 4 presents the main results. Finally, Section 5 summarises the findings and suggests future directions for research.

## 2. State-of-the-Art

A number of scientific works have been dedicated to predicting the efficient use of solar panels and the electricity they generate due to the fact that new energy sources are increasingly important for our contemporary society. This section summarises the research carried out over the past few years and the results achieved.

Kim et al. [22] propose a model for predicting the solar power, obtained from Photovoltaic (PV) panels and for optimising the tilt angle in the case of Daegu city in South Korea. For this purpose, the authors apply several Machine Learning algorithms like: Linear Regression (LR), Random Forest (RF), Gradient Boosting (GB), Support Vector Machine (SVM), Least Absolute Shrinkage and Selection Operator (LASSO), and also consider several influential factors (weather conditions, availability of dust and aerosol).

It has been proven that such an approach leads to increased effectiveness at energy production. Wei [23] investigates how to improve the functionality of PV systems, which are located in Tainan City, Taiwan. Solar radiation of the panels' surface at various tilt angles is predicted via the utilisation of four Machine Learning (ML) algorithms: LR, RF, Multilayer Perceptron (MLP) and K-nearest Neighbors (kNN). The optimal value of the solar panels' tilt angel is also found. Machine learning techniques related to the construction of Artificial

Neural Networks (ANNs) are used by Kamal et al. [24] to reconfigure the topology of PV arrays and to achieve their optimal workability .

The findings point out that the presented mechanism with very high accuracy is capable of to outlining the best topologies (among the following: series parallel, parallel, bridge link, honeycomb, and total cross tied) for the PV panels' deployment. Dependence between the correct PV panels' installation and their efficiency is investigated by Kim and Byun [25] as the authors predict power generation. The XGBoost algorithm is applied for solving a regression task and to give a very accurate prognosis regarding the electricity generation. Khilar et al. [26] propose a model, based on the deep belief network, for detecting the dust level on solar panels. Such an investigation is important for the places where there is almost no wind and rain, and at same time, the PV system must work efficiently. The considered input variables include dust particles on panels, temperature, and solar irradiance, which are important for identifying the frequency of the manual or automatic cleaning of the panels.

Khan et al. [27] rely on ensemble ML algorithms RF, XGBoost, and catboost to find the optimal direction for the placement of PV panels. The proposed approach can predict solar power at two levels: at the first level, base models are created via the utilisation of XGBoost and catboost, and the resulted predictions are used at the second level where RF is applied for building a metamodel. The task of this metamodel is to learn what is the best way to use the predictions gathered from the base models. The presented method is evaluated and compared with other ML algorithms and it is proved its better performance is proven. Gautam M et al. [28] show a framework to maximise the usage of solar power, which is built on the Decision Tree (DT) algorithm. The idea behind it is to find a strategy for switching solar and grid systems, which are connected to a common node, to perform efficient energy management. This, DT predicts the switching configuration and, in this way, the cost for electricity is reduced at an increased consumption.

Predicting the usage of solar power in summer and winter in Mashhad, Iran is discussed by Almadhor et al. [29]. This investigation is conducted in the context of the realisation of smart cities and through stimulating the citizens to used renewable energy. The prediction is performed via constructed ANN, which solves a linear regression problem. Shaaban et al. [30] use the Machine Learning approach to evaluate the dust volume on solar Photovoltaic panels as dust contributes to decreasing the generated energy. The proposed model, based on a regression tree, is compared with an ANN model in order to be demonstrated its high performance. When a threshold value of the dust level is reached, a cleaning procedure is triggered. Bulusu et al. [31] propose a predictive model that points out the hourly energy production from solar microgrids. This novel approach uses ANNs and includes two parts: (1) for features extraction and (2) for predicting the energy production via the created models for the given hours. The authors recommend not using data older than two years to train the models, as solar panels degrade over time and significant differences in the training and testing data can occure. An investigation related to the performance of PV grid-based systems is presented by Yar et al. [32] as the analysis relies on several methods like: conducting experiments, accomplishing simulations, and applying Machine Learning. The Logistical Regression technique is utilised to evaluate the difference in the performance between simulation results and real-time systems. Then, the findings obtained regarding monoperk and polyperk crystalline systems as well as their advantages and bottlenecks are discussed.

Mahesh et al. [33] propose a novel method for evaluating the efficiency of PV panel systems. It combines the Maximum Power Point Tracking (MPPT) technique and SVM Machine Learning to predict the maximal value of the power generated by a PV solar panel. The presented method is compared with the available ones (ANN, fuzzy logic, perturb and observe, and incremental conductance) and its high performance is proven. The problem regarding how power effective the PV panels placed on the building's façade are examined by Vahdatikhaki et el. [34]. Surrogate modeling is applied to simulate solar radiation as the last one is predicted via the RF algorithm in three scenarios. Then, an optimisation

procedure is applied regarding the obtained RF model hyperparameters via the usage of the genetic algorithm. The authors conclude that such an approach possesses big potential to simulate and predict with high accuracy the solar radiation of vertically placed PV panels on the buildings' surface.

Pasion et al. [21] uses Machine Learning techniques to construct models based on data from twelve sites in the United States of America to predict Photovoltaic energy production without irradiance data. Incorporating irradiance data into solar energy forecasts poses challenges in terms of data accuracy, computational complexity, maintenance costs, lower interpretability and limitations in regions with sparse historical data. The study uses readily available parameters such as location, time and weather conditions. By comparing six Machine Learning algorithms, including deep learning and ensemble models, the study identified distributed random forest as the most effective. The study also found that ambient temperature, humidity and cloud cover were the most important variables for prediction. The authors highlight the possibility of precise predictions even without irradiance data.

## 3. Materials and Methods

In this section, the materials and methods used in this project are described.

### 3.1. Data Set Description

This work uses the Pasion et al. location data [21,35], which contains data collected from twelve Department of Defense (DoD) solar installations at different locations within the United States of America (USA) from the year 2017 to 2018.

The information is stored in an unique CVS file with 21,046 rows and 17 columns. The attributes included are included in the following, along with brief descriptions:

- **GENERATION**
  - **PolyPwr:** Solar power generation in Watts.
- **TEMPORAL**
  - **Date:** Date of measurement.
  - **Time:** Time of measurement.
  - **Month:** Month of measurement.
  - **Hour:** Hour of measurement.
  - **YRMODAHRMI:** Combination of year, month, day, hour and minute.
  - **Season:** Season of the year.
- **GEOGRAPHICAL**
  - **Location:** Geographic location of measurement site.
  - **Latitude:** Latitude coordinates in degrees.
  - **Longitude:** Longitude coordinates in degrees.
  - **Altitude:** Altitude in meters.
  - **Pressure:** Atmospheric pressure in millibars.
- **METEOROLOGICAL**
  - **Humidity:** Humidity level in percentage.
  - **AmbientTemp:** Ambient temperature in Celsius.
  - **Wind.Speed:** Wind speed in km/h.
  - **Visibility:** Visibility distance in km.
  - **Cloud.Ceiling:** Height of cloud cover in km.

The sampling period is highly variable throughout both the date and location, but ranges from 15 min to hours.

### 3.2. Data Preprocesing

For the experiments, the original database, which had already been pre-processed, was used. The authors of the baseline study [21], filtered the database to cover exclusively

cover the time window from 10:00 a.m. to 3:45 p.m. This was in order to avoid modelling periods of darkness and low sunlight, also reducing losses caused by shadows. The variable "YRMODAHRMI" mentioned above is dropped in this stage as it does not bring new information. There were not missing values within the database.

In this study, the categorical variable "Season" is used after one-hot encoding, depending on the model used, and normalisation of the data is conducted. From now on this dataset is called DB1.

### 3.3. Data Visualization

In the context of this work, it is useful to identify the relationships between features and their effect on the variable to be predicted, which is "PolyPwr" for DB1. In this section, the visual analysis for the database is provided.

In Figure 1, all locations of this study are displayed on a map, along with their names and circles of different colours and sizes, representing the number of instances in DB1 for each place. In addition, the accompanying graph on the right side of the figure provides information about the colour coding and its corresponding frequencies.

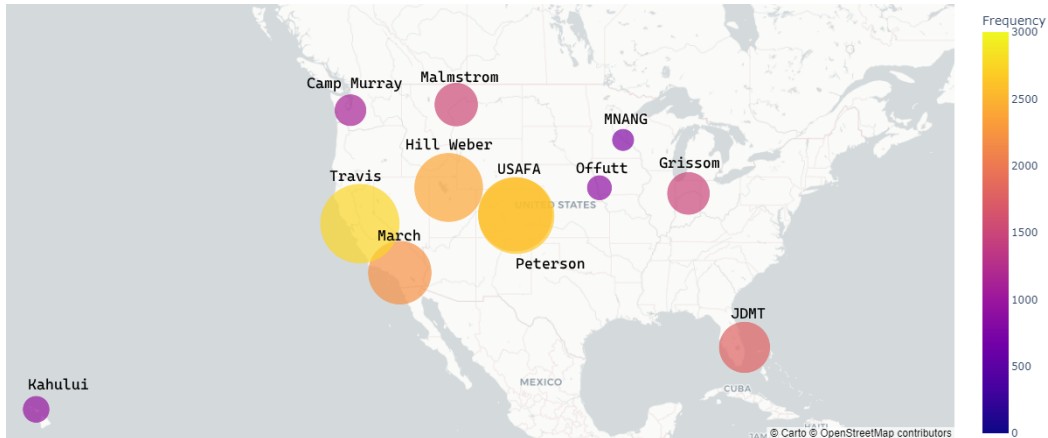

**Figure 1.** Geographic locations of the studied sites (This map was generated using the Python package plotly.io).

The larger the circle, the more data there is for that place. For example, the number of instances in Kahului is substantially smaller than in Travis, and it is reflected in their size and colour difference.

Finally, it can be seen that for Peterson and USAFA there is an overlap of their circles, since they are geographically very close as well as have a similar number of instances, therefore making it difficult to distinguish one from the other.

In Table 1, descriptive statistics for numeric variables can be found. The columns describe the number of instances (count), their mean ($\overline{x}$), their standard deviation (std), their lowest value (min), their highest value (max), and, the 25th, 50th, and 75th percentiles.

The representation of the variable to be predicted, Power Generation, is presented in Figure 2, which displays the distribution of the different values this variable assumes. It is evident that sufficient representation is provided from 0 to 23. Beyond this range, however, the same statement cannot be made.

**Table 1.** Descriptive statistics of numerical variables in DB1. $\bar{x}$ represents the arithmetic mean of the sample.

| Variable | Count | $\bar{x}$ | *Std* | min | 25% | Median | 75% | max |
|---|---|---|---|---|---|---|---|---|
| Latitude | 21045 | 38.2 | 6.3 | 20.9 | 38.2 | 39 | 41.1 | 47.5 |
| Longitude | 21045 | −108.6 | 16.4 | −156.4 | −117.3 | −111.2 | −104.7 | −80.1 |
| Altitude | 21045 | 798.8 | 770.7 | 1 | 2 | 458 | 1370 | 1947 |
| Month | 21045 | 6.6 | 3 | 1 | 4 | 7 | 9 | 12 |
| Hour | 21045 | 12.6 | 1.7 | 10 | 11 | 13 | 14 | 15 |
| Humidity | 21045 | 37.1 | 23.8 | 0 | 17.5 | 33.1 | 52.6 | 100 |
| AmbientTemp | 21045 | 29.3 | 12.4 | −20 | 21.9 | 30.3 | 37.5 | 65.7 |
| PolyPwr | 21045 | 13 | 7.1 | 0.3 | 6.4 | 13.8 | 18.9 | 34.3 |
| Wind.Speed | 21045 | 10.3 | 6.4 | 0 | 6 | 9 | 14 | 49 |
| Visibility | 21045 | 9.7 | 1.4 | 0 | 10 | 10 | 10 | 10 |
| Pressure | 21045 | 925.9 | 85.2 | 781.7 | 845.5 | 961.1 | 1008.9 | 1029.5 |
| Cloud.Ceiling | 21045 | 516 | 301.9 | 0 | 140 | 722 | 722 | 722 |

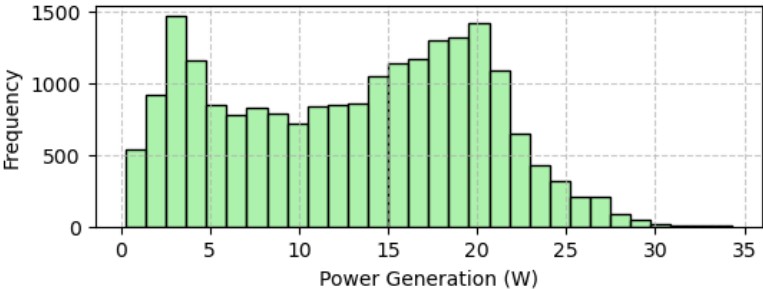

**Figure 2.** Distribution of power generation.

In Figure 3, the correlation between numerical variables in DB1 are shown. Most notable, the relationship between Pressure and Altitude is noted, having an almost a perfect negative correlation, which is consistent with their expected relationship.

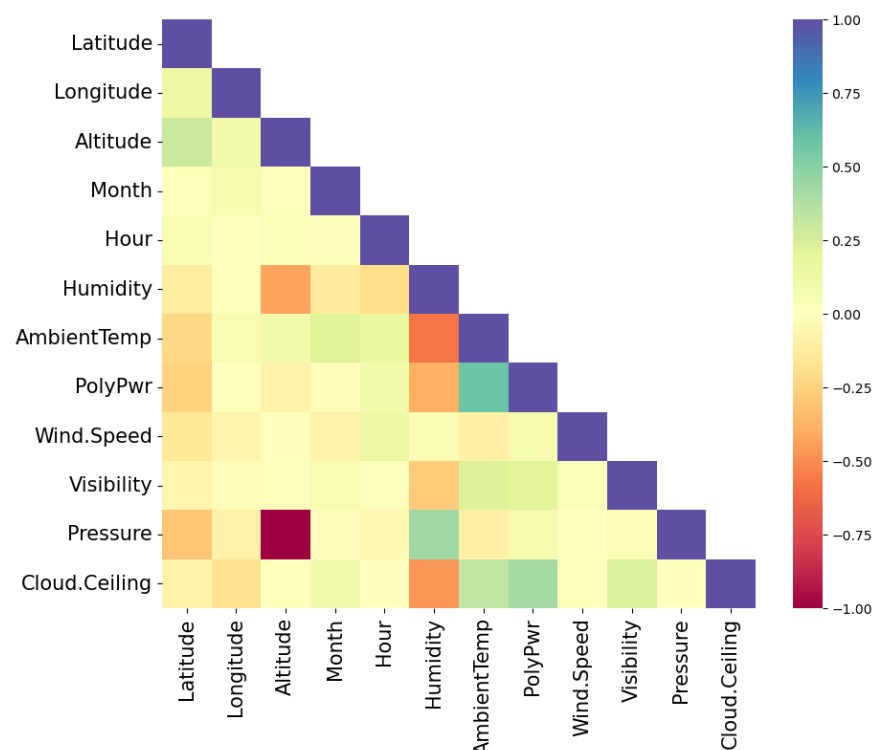

**Figure 3.** Correlation matrix of all numerical variables within the database.

A detailed view of the correlation between PolyPwr and the rest of the numerical variables is shown in Figure 4. The graph shows that it has a high positive correlation with ambient temperature and cloud cover, and a negative correlation with humidity and latitude. In particular, there appears to be almost no correlation with longitude and month.

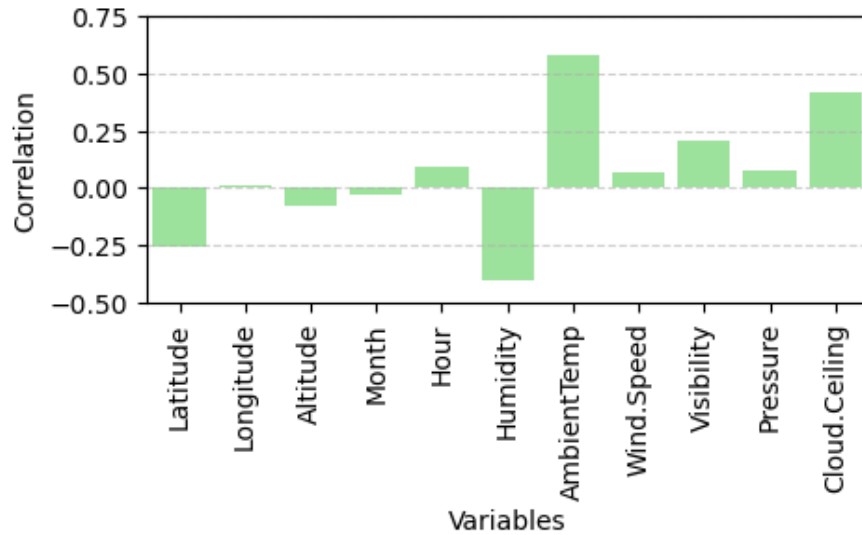

**Figure 4.** Correlation of all numerical variables in the database with the variable PolyPwr.

### 3.4. Input Parameters Description

The primary objective of this study is to develop accurate predictions of the energy self-supply from Photovoltaic generators. In order to do so, the study uses temporal, geographical, and meteorological features.

In the work of Pasion et al. [21], the variables used were latitude, month, hour, ambient temperature, pressure, humidity, wind speed, cloud ceiling, and visibility.

In the present study, however, the variables altitude and season are used as well.

### 3.5. Regressors

In this section, the importance and use of regressors in the experiments are explained. Likewise, the process of model building is generally outlined and the parameterisation chosen for each specific regressor is detailed.

The type of regressor chosen for the model plays a key role in the final result to be obtained. Here, regressors are used to model the relationship between the solar energy production and the temporal, meteorological, and geographical data, allowing us to optimise the accuracy of both real-time and hourly predictions and facilitate solar energy monitoring and design for practical applications.

In the baseline study [21,35], H2O AutoML was used. The H2O module is "an open source, in-memory, distributed, fast, and scalable Machine Learning and predictive analytics platform that allows you to build Machine Learning models on big data and provides easy productionalisation of those models in an enterprise environment", according to its own documentation [36]. There, six different regressors, with their default input values, were used and compared. The best results were obtained using the distributed random forest regression algorithm, so our comparisons will be made exclusively with their implementation of this model.

For this experimentation, the library scikit-learn is used for the models of Linear Regression and Random Forest. For the Artificial Neural Network, Tensorflow and Keras were used.

Random Forest

In the application of this work, the best results are obtained using the following parameterisations for RF:

- The number of trees was set to {50, 100, 250, 500}.
- To continue growing each tree at the training stage, a minimum number of instances was required per split, 4, and also per leaf, 2.
- The maximum features considered for splitting a node were the square root of the total number of features (*sqrt*).
- The maximum depth of each tree was limited to 22.

### 3.6. Experimental Pipeline

This section describes the methods used in order to perform both the cross validation, validation, and evaluation of the models. In addition to do this, all models were tested on five different seeds.

This is a key in guaranteeing both the reliability of the predictions and the models ability to handle unknown data.

### 3.6.1. Validation Methods

The cross validation used in this project allows us to go through several subsets of the main dataset in a systematic way, testing the model in different scenarios. Its use is described below:

All the Database

In this approach, the entire dataset is used for validation, where k-fold cross validation is used with a k of 5. Essentially, training is performed using four subsets of the five from the entire dataset, while the remaining one is used for testing. This is repeated k times, in this case five, so that all subsets are used for validation.

By Location

For this case, the same cross-validation technique is used as in the previous case, however, a separate model is trained and tested for each individual location.

Taking One Location Out

In this case, the data from one location is omitted from the training process, while the rest of the locations are used in their entirety. The performance of the model with the omitted data is then evaluated. This allows us to see the model's ability to generalise it to unknown scenarios.

Taking One Location Out but Leaving One Week

In the last approach, a whole location is not excluded from training, instead, information from the first week is retained while excluding the rest of its time horizon. For the remaining locations, all instances are used. Afterwards, the same procedure is repeated while including the location details that were omitted during the model learning phase for evaluation.

Then, the same procedure as before is performed, using the omitted location information that has not been fed in the model learning process for its evaluation.

This gives us insight on how providing contextual 1 week temporal data can improve or worsen the working model.

### 3.6.2. Evaluation Methods

In this section, the evaluation metrics used to assess the effectiveness of the models are discussed.

R-Squared ($R^2$)

The coefficient of determination, often known as R-squared, is the proportion of the total variance of the dependent variable that can be explained by the estimated regression model [37].

A value close to 1 means that the model is able to explain a large part of the variation in the data, lower values mean that it lacks this ability and therefore does not fit the data well. It is calculated using the following formula:

$$R^2 = 1 - \frac{SS_{\text{res}}}{SS_{\text{tot}}}$$

where

$SS_{\text{res}}$: Sum of squared differences between predicted and actual values
$SS_{\text{tot}}$: Total sum of squared differences between actual values and their means.

By this definition, we can see that if $SS_{\text{res}}$ is less than $SS_{\text{tot}}$, this means that the predictions are worse than a model that simply predicts the median of the true values and that $R^2$ will take a negative value.

CO2 Emissions

To assess the climate impact, calculations were performed to measure the carbon emissions generated during both the training and testing processes of various models based on Random Forest (RF). These emissions, quantified in grams of carbon dioxide equivalent (g$CO_2$eq), are attributed to the cloud or the personal computing resources used for code execution.

All the experiments described were implemented locally in Spain. As of 2022, the carbon efficiency in the country was 0.163 kg$CO_2$eq/kWh according to Carbon Footprint Ltd. [38].

The equipment used to perform this work had an Intel Core i5-10300H CPU of 2.50 GHz, 8 Gb of RAM, and a NVIDIA GeForce GTX 1650 Ti GPU and all estimations were calculated with the CodeCarbon emissions tracker version 222 [39], thanks to the previous work of Lacoste et al. [40] and Lottick et al. [41].

## 4. Results

As seen in the previous section, the performance of the algorithms has been evaluated on the basis of four different metrics, but for the analysis of the results, R-squared and the total emissions estimated in g$CO_2$eq are used as comparison metrics. Only the most relevant models are presented below and their results consist on the average of the folds over five different seeds.

### 4.1. Results Related to Objective One

As mentioned in the introduction, the results of Pasion et al. are reproduced. In particular, the instantaneous prediction model proposed in the study is developed and applied, and the state of the art is extended.

### 4.1.1. All the Database

Using the methodology of Pasion et al, their results were successfully reproduced with minimal variation. Figure 5 displays the $R^2$ values obtained from this experiment, illustrating two model variations that differ only in the number of estimators used, i.e., the number that accompanies their name.

Thanks to this, we can visually compare the difference between both versions. As the number of estimators increases, the model's fit improves. However, this enhancement results in higher emissions due to the need for extended runtime.

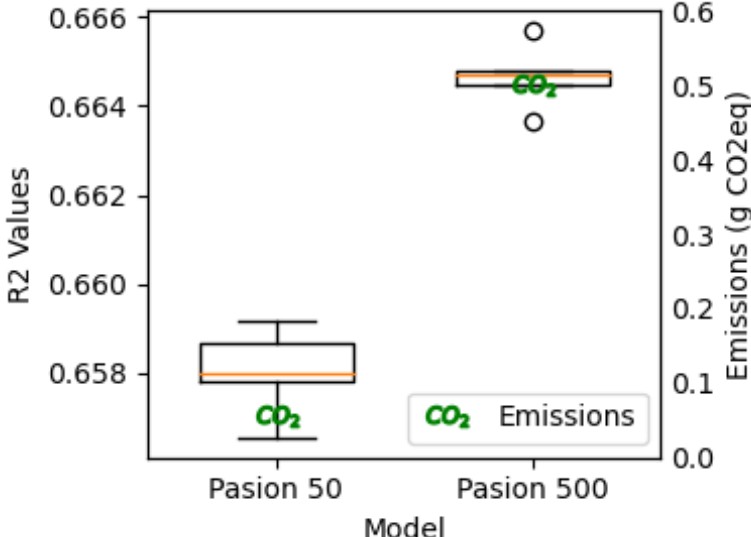

**Figure 5.** Comparative graph of the models proposed by Pasion et al.

4.1.2. By Location

A fair comparison of the results cannot be made because the cross-validation values using location modelling were not provided in the original study. However, Figure 6 displays the results using 50 estimators. The emissions generated to train these 12 models were 0.028 *gCO₂eq*.

In the box plot, we observe that the model is capable of fine-tuning itself greatly for certain places such as Camp Murray and Travis, reaching values close to 0.8. On the other hand, Kahului, JDMT, and USAFA stand out as having obtained the worst results, with the worst-case obtaining values lower than 0.4, well below the mean.

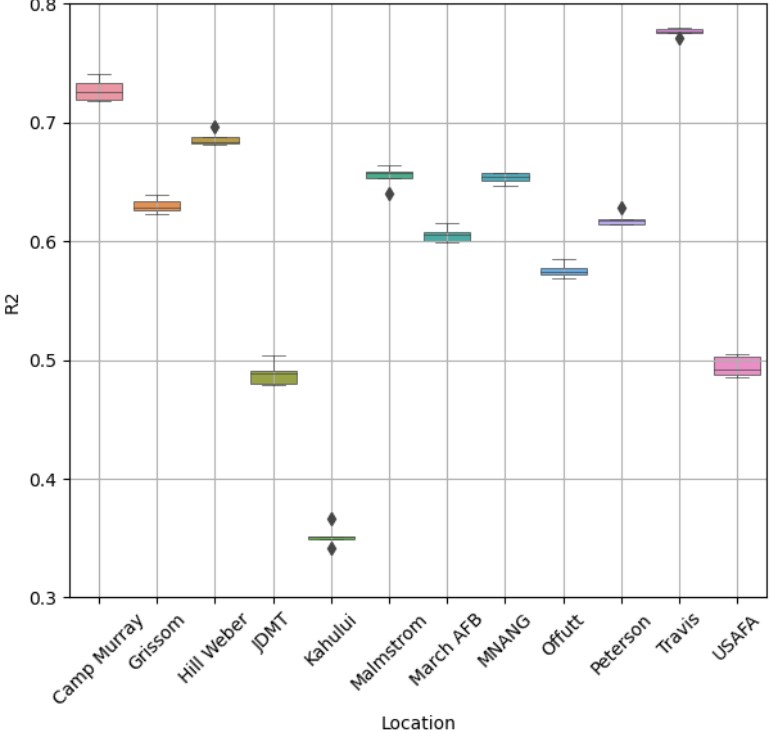

**Figure 6.** Results obtained from location modelling using the parametrisation proposed by Pasion et al.

*4.2. Results Related to Objective Two*

To enhance the precision of PV generation prediction, a distinct approach is suggested. Various hyperparameterisations of the algorithm are analysed, and different folding techniques are compared with the previous methods mentioned earlier.

### 4.2.1. All the Database

In Figure 7, three different implementations of the model proposed in our experiment are displayed. These results are obtained using the Random Forest algorithm and the parameterisations mentioned above. As depicted, each implementation is associated with a specific number of estimators, indicated next to the implementation name.

The most impressive outcomes are obtained using the model RF 1-500, followed by RF 1-100 and RF 1-50. This pattern shows a consistent relationship between the number of estimators used and larger $R^2$ results, reinforcing our earlier observations.

Alongside, the results of the base experiments are presented. Notably, our results surpassed those of the base study conducted by Pasion et al. [21]. It can be observed that our method not only achieves enhanced $R^2$ values but also a significant reduction in the quantity of emissions generated. This improvement further stresses the effectiveness of our approach in achieving both greater predictive accuracy and environmental sustainability.

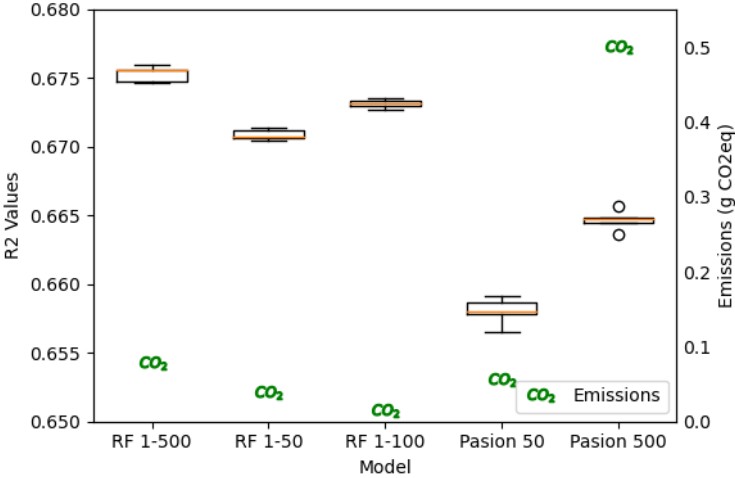

**Figure 7.** Comparison between results obtained from modelling using the parametrisation proposed by this study and by Pasion et al.

### 4.2.2. By Location

In Figure 8, the results are shown when using our model with random forest and 250 estimators, with which we obtain a better $R^2$ and total emissions of 0.074 $gCO_2eq$. However, similar results were obtained with 50 and 100 estimators, with emissions of 0.026 $gCO_2eq$, which is lower than the original.

### 4.2.3. Taking One Location Out

Figure 9 shows that omitting a site from the training leads to significantly worse results, including negative values of $R^2$, as observed for JDMT. This effect could be attributed to the differences in climate between the locations. The employed model was the RF model with 250 estimators and generated 0.126 $gCO_2eq$ emissions.

### 4.2.4. Taking One Location Out but Leaving One Week

By giving the model a week of context, it can be seen in Figure 10 that even using a very small time horizon compared to the rest of the data, all models are improved and none of them have negative results anymore. The model used was the RF model with 250 estimators, which produced 0.171 $gCO_2eq$ emissions.

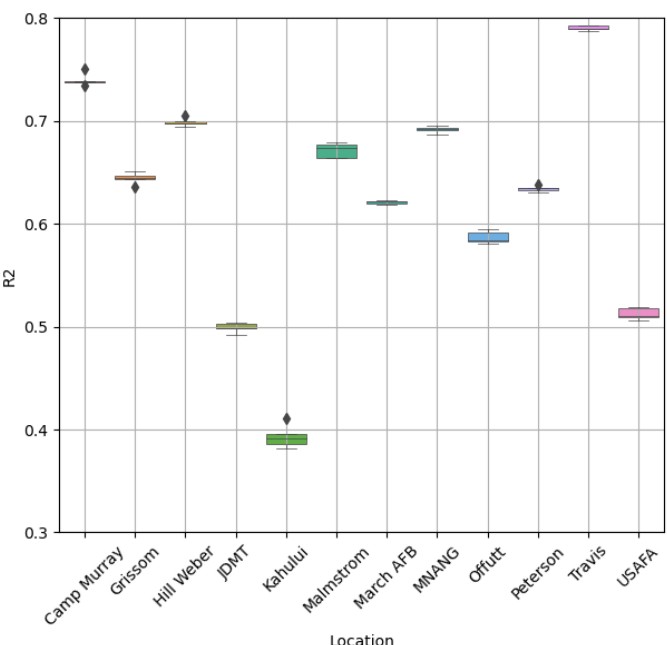

**Figure 8.** Results obtained from location modelling using the parametrisation proposed by this study.

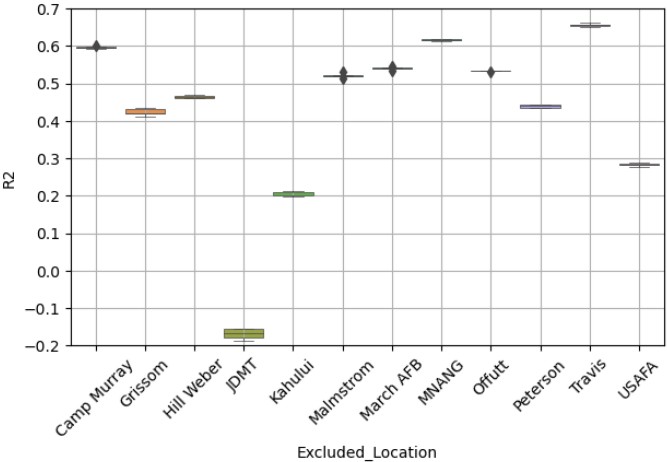

**Figure 9.** Results from location modelling excluding a location using our proposed model.

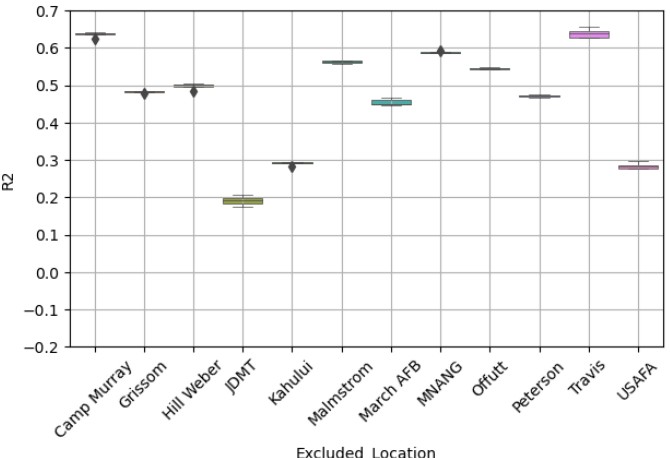

**Figure 10.** Results from location modelling excluding a location and providing 1-week data context using our proposed model.

## 5. Discussion and Conclusions

In this work, different models have been proposed with the aim of optimising the prediction of the energy generated by Photovoltaic installations. Not only the values obtained using the method proposed by Pasion et al. were improved, but also the number of emissions was decreased.

When modelling by location, prominent differences were observed depending on the location, with both very high and very low values of accuracy. Omitting a location from the training showed how all models worsened, reaching negative $R^2$ results, but then giving a week of data from the location in the form of context made all the models improve, demonstrating that only a small time horizon has an impact, achieving better results.

This is due to the strong dependence of Photovoltaics on environmental factors, but especially on geographical factors, where two installations at the same latitude but one at the sea level and the other at a high altitude will behave differently even on the same day; the same is true for sites at different latitudes where the difference between the seasons is more noticeable.

On the other hand, it has been found that across the different experiments, as a general rule, using the same parameterisation, the higher the number of estimators, the better the prediction results, but that this increases the carbon emissions generated by the model.

Therefore, we conclude that, within the scope of this research, it was not possible to make a single model that can be adapted to all the sites studied, due to the considerable variations in the different variables. Prediction models should be designed considering the individual characteristics of the sites where they are to be implemented, thus improving the prediction results and allowing for a more efficient management of renewable energy resources.

Finally, it is important to include environmental impacts in the process of developing and using models. While they may seem marginal in this case, they are emissions that are emitted over the entire life cycle of the model. This approach allows us to create reliable models while managing all of our resources in a sustainable and environmentally responsible manner, so we believe that minimising and quantifying these emissions is an important angle to include in all model development.

**Author Contributions:** Conceptualization, A.P.A.-C., A.E. and J.D.N.-G.; methodology, A.P.A.-C., A.E. and J.D.N.-G.; software, A.P.A.-C., A.E. and J.D.N.-G.; validation,A.P.A.-C., A.E. and J.D.N.-G.; formal analysis, A.P.A.-C., A.E. and J.D.N.-G.; investigation, A.P.A.-C., A.E. and J.D.N.-G.; resources, A.P.A.-C., A.E. and J.D.N.-G.; data curation, A.P.A.-C., A.E. and J.D.N.-G.; writing—original draft preparation, A.P.A.-C., A.E., J.D.N.-G. and M.I.; writing—review and editing, A.P.A.-C., A.E., J.D.N.-G. and M.I.; visualization, A.P.A.-C., A.E. and J.D.N.-G.; supervision, A.E. and J.D.N.-G. ; project administration, A.E., J.D.N.-G. and M.I.; funding acquisition, M.I.. All authors have read and agreed to the published version of the manuscript.

**Funding:** This research is supported by the Bulgarian National Science Fund in the scope of the project "Exploration the application of statistics and machine learning in electronics" under contract number $\kappa\pi$-06-H42/1.

**Institutional Review Board Statement:** Not applicable.

**Data Availability Statement:** Data available on request.

**Conflicts of Interest:** The authors declare no conflict of interest.

## Abbreviations

The following abbreviations are used in this manuscript:

| | |
|---|---|
| PV | photovoltaic |
| LR | linear regression |
| RF | random forest |
| GB | gradient boosting |

| SVM | support vector machine |
| LASSO | least absolute shrinkage and selection operator |
| ML | machine learning |
| ANNs | artificial neural networks |
| DT | decision tree |
| MPPT | maximum power point tracking |

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
