# Peer review of "Development of AI-Based Tools for Power Generation Prediction"

_computation, doi:10.3390/computation11110232_

Round 1
Reviewer 1 Report
Comments and Suggestions for Authors
The authors proposed various models for predicting photovoltaic power generation based on meteorological, temporal, and geographical variables, without using irradiation values. The article is interesting. However, this reviewer has the following concerns on the submitted manuscript:
· The article is very roughly written and not well-formatted and well-organized.
· The background of the study should be enhanced by adding recent and relevant literature. What are the drawbacks of the existing literature and how are you tackling those challenges?
· Please rename section 2 as 'State-of-the-art' not 'State of art'.
· Reference numbers for Kim et al., Gautam M et al., Khan et al., and many more?
What is 'Me' on Table 1?
· No need to define the evaluation metrics and their equations. The selected metrics of Section 3.6.2 are very well established. Just adding their sources are enough.
· Please provide a comparative study with the existing techniques. It is not clear how your model is performing better that other models.
· A conclusion section is needed.
Overall, the article requires significant modifications before acceptance.
Comments on the Quality of English LanguageExtensive editing of English language required
Author Response
Dear Reviewer, Thank you so much for your comments! We are very grateful!

Reviewer 2 Report
Comments and Suggestions for Authors
[1] gCO2eq?
[2] The first part of the paper is not described in enough detail. For example, different machine learning algorithms were used, which ones exactly?
[3] There should be a lineage or logic in the related research work section. It should be described in a general structure.
[4] The general objectives of this work should be placed in the first section.
[5] The organization of the chapters of the paper should be placed at the end of Part I. The overall organization of the paper is confusing. The overall organization of the paper is confusing.
[6] The formulas of the corresponding methods should be used in the thesis instead of utilizing their python code presentation. Especially in chapter 3.5.1.
[7] What methods are being compared in the paper should be described in detail.
[8] Why these methods were compared should also be described.
[9] What the results described in the paper suggest should be strengthened.
[10] Abbreviations should be written in a consistent format. Some times it is capitalized and some times it is lower case. It is not right.
[11] Papers three years old should be avoided as references.
[12] References should be at least 40.
[13] Try to use higher level papers as references.
Author Response

(The authors gave the same response as above.)

Reviewer 3 Report
Comments and Suggestions for Authors
This manuscript studied different machine learning algorithms and techniques and developed a model that can predict photovoltaic generation. Building on previous work by Pasion et al. [4], an algorithm is developed that predicts the energy generated by photovoltaic installations. Both validation methods: cross-validation and evaluation metrics are used.
In my opinion, the manuscript offers a good contribution. The manuscript is well-written and well-presented. The references are relevant and up to date. However, to improve the methodology and presentation of the manuscript, the authors should consider the following comments:
1. The "Abstract and Discussion" sections need improvement, the authors should rewrite these sections.
2. Authors must add the source of figure 1.
3. Define each abbreviation that is not defined in the paper.
4. In the introduction section, the authors should add more studies related to this paper.
5. In MSE and MAE equations, ypred and ytrue symbols are not defined. Please correct this.
6. Add a "conclusion" section at the end of the paper. To make more sense, the concluding section should be expanded. In conclusion, the study's contributions should be emphasized. What distinguishes this research from earlier research? In what ways does this article contribute to the field's understanding? At the conclusion of this work, the limitation should be mentioned.
7. The authors should put the dataset used of this study in the “Supporting Information File".
==============================================
Author Response

(The authors gave the same response as above.)

Round 2
Reviewer 1 Report
Comments and Suggestions for Authors
Please accept
Comments on the Quality of English LanguageMinor editing of English language required
Reviewer 2 Report
Comments and Suggestions for Authors
I have no more comments.